# Does Breast Surgery Type Alter Incidental Axillary Irradiation? A Dosimetric Analysis of the “Sentinel Envahi et Randomisation du Curage” SERC Trial

**DOI:** 10.3390/cancers16061198

**Published:** 2024-03-19

**Authors:** Camille Nicolas, Claire Petit, Agnès Tallet, Jean-Marie Boher, Leonel Varela Cagetti, Veronique Favrel, Laurence Gonzague Casabianca, Morgan Guenole, Hugues Mailleux, Julien Darreon, Marie Bannier, Monique Cohen, Laura Sabiani, Camille Tallet, Charlene Teyssandier, Anthony Gonçalves, Alexandre De Nonneville, Leonor Lopez Almeida, Nathan Coste, Marguerite Tyran, Gilles Houvenaeghel

**Affiliations:** 1Department of Radiotherapy, Institut Paoli-Calmettes, 13009 Marseille, France; petitc2@ipc.unicancer.fr (C.P.); richarda@ipc.unicancer.fr (A.T.); varelacagettil@ipc.unicancer.fr (L.V.C.); favrelv@ipc.unicancer.fr (V.F.); gonzaguel@ipc.unicancer.fr (L.G.C.); guenolem@ipc.unicancer.fr (M.G.); tyranm@ipc.unicancer.fr (M.T.); 2Biostatistics and Methodology Unit, Institut Paoli-Calmettes, INSERM (National Institute of Health and Medical Research), IRD (Development Research Institute), Aix Marseille University, 13009 Marseille, France; boherj@ipc.unicancer.fr; 3Department of Medical Physic, Institut Paoli-Calmettes, 13009 Marseille, France; mailleuxh@ipc.unicancer.fr (H.M.); darreonj@ipc.unicancer.fr (J.D.); 4Department of Surgical Oncology, Institut Paoli-Calmettes, CRCM (Research Cancer Centre of Marseille), 13009 Marseille, France; bannierm@ipc.unicancer.fr (M.B.); cohenm@ipc.unicancer.fr (M.C.); sabianil@ipc.unicancer.fr (L.S.); 5Faculty of Medical and Paramedical Sciences, Medical School, Aix-Marseille University, 13005 Marseille, France; camille.tallet@etu.univ-amu.fr (C.T.); charlene.teyssandier@etu.univ-amu.fr (C.T.); 6Department of Medical Oncology, CNRS (National Center of Scientific Research), INSERM (National Institute of Health and Medical Research), Institut Paoli-Calmettes, CRCM (Research Cancer Centre of Marseille), Aix-Marseille University, 13009 Marseille, France; goncalvesa@ipc.unicancer.fr (A.G.); tassindenonnevillea@ipc.unicancer.fr (A.D.N.); 7Department of Clinical Research, Institut Paoli-Calmettes, 13009 Marseille, France; lopezalmeidal@ipc.unicancer.fr (L.L.A.); costen@ipc.unicancer.fr (N.C.); 8Department of Surgical Oncology, CNRS (National Center of Scientific Research), INSERM (National Institute of Health and Medical Research), Institut Paoli-Calmettes, CRCM (Research Cancer Centre of Marseille), Aix-Marseille University, 13009 Marseille, France; houvenaeghelg@ipc.unicancer.fr

**Keywords:** breast cancer, radiotherapy, axillary lymph nodes, mastectomy

## Abstract

**Simple Summary:**

Currently, breast surgery type directly affects the management of the axillary fossa in patients with breast cancer. Trials that validated the omission of completion axillary dissections after a positive sentinel lymph node biopsy had excluded patients treated with a total mastectomy. Radiotherapy and its incidental axillary dose seem to be a major element of local axillary control in this setting, but these are only described in the literature after conservative surgeries. Within a sub-study of an SERC trial, which included patients treated with a total mastectomy, we demonstrated that the axillary dose delivered via radiotherapy is of the same level as that already known after conservative surgery. This element could contribute to the data allowing us to dispense with completion axillary lymph node dissections in the case of a positive sentinel lymph node biopsy after a total mastectomy.

**Abstract:**

Background. An incidental axillary dose of adjuvant radiotherapy using tangential beams is usually given after breast-conserving surgery for breast cancer. The goal of this sub-study was to evaluate this incidental dose in the setting of post-mastectomy radiotherapy (PMRT) according to two different radiotherapy techniques. Methods. Patients participating in a randomized SERC trial who received PMRT in a single center were included. We collected the incidental axillary dose delivered to the Berg level 1 using different dosimetric parameters and compared two techniques using Student’s *t*-test: three-dimensional conformal radiotherapy (3D-CRT) and volumetric arc therapy (VMAT). Results. We analyzed radiotherapy plans from 52 patients who received PMRT from 2012 to 2021. The mean dose delivered to the Berg level 1 was 37.2 Gy. It was significantly higher with VMAT than with 3D-CRT—43.6 Gy (SD = 3.1 Gy) versus 34.8 Gy (SD = 8.6 Gy) *p* < 0.001. Eighty-four percent of the Berg level 1 was covered by 40 Gy isodose in the VMAT group versus 55.5% in the 3D-CRT group *p* < 0.001. Conclusions. On the Berg level 1, PMRT gives a dose at least equivalent to the one given by post-breast-conserving surgery radiotherapy, making it possible to limit completion axillary lymph node dissections in select pN1a patients treated with a mastectomy. Modern radiotherapy techniques like VMAT tend to increase this incidental dose.

## 1. Introduction

Breast-conserving surgery (BCS) has become the preferred strategy in treating early breast cancer since it allows for the same rate of overall survival as radical surgery, albeit with cosmetic benefits. However, a mastectomy remains required in some situations, such as a high tumor/breast ratio, multicentric tumors, extensive intraductal component, BRCA mutation, or the patient’s preference.

Axillary lymph node assessment has considerably progressed during the last twenty years, shifting from axillary lymph node dissection (ALND) to sentinel lymph node biopsy (SLNB), followed by ALND if positive and then proceeding to ALND omission in selected early breast cancer patients with positive sentinel nodes. Three randomized trials have validated ALND omission in T1-2N0, pN1mi, or pN1a (limited to two macroscopically positive lymph nodes) patients treated with BCS and SLNB, including two trials with a ten-year follow-up [1,2,3]. In these three trials, almost all the patients received adjuvant radiotherapy to the breast and systemic therapy. Although the rate of positive non-sentinel lymph nodes assessed in the ALND arm for pN1a patients was 27.3%, the 10-year ipsilateral axillary recurrence rate in the SLNB arm was only 1.5% in the ACOSOG Z0011 trial [4]. The authors suggested that adjuvant therapies likely accounted for this high rate of regional control despite ALND avoidance, although the effects of radiotherapy and systemic treatment could not be differentiated. However, in a retrospective study comparing axillary recurrence rates in BCS pN0 patients treated with either whole-breast radiotherapy or intraoperative electron therapy, Gentilini et al. found that whole-breast radiotherapy prevents axillary recurrence (with a 10-year cumulative incidence of axillary recurrence of 1.3% versus 4%, respectively, *p* < 0.001) [5]. These results support the hypothesis that the incidental irradiation of the axillary fossa by tangential beams may reduce axillary recurrence.

Patients enrolled in the IBCSG 23-01 trial underwent an upfront mastectomy in 9% of cases, while only BCS was allowed in the ACOSOG Z0011 trial [1,2]. Thus, there is a lack of data concerning axillary surgery de-escalation as concerns upfront mastectomy. In the AMAROS trial, voluntary axillary radiotherapy was non-inferior to ALND for axillary recurrence and overall survival with 10 years follow-up [6]. Completing axillary treatment through ALND or axillary-directed radiotherapy is recommended after a mastectomy and one or two positive nodes detected on a SLNB [7,8,9].

We usually indicate the chest wall with regional nodal irradiation after a mastectomy in the case of positive axillary sentinel lymph nodes, and in this situation, adjuvant systemic therapy is often recommended. Therefore, by combining these therapies, the omission of any further axillary treatment could also be applied to patients undergoing mastectomy, provided they meet the other ACOSOG Z0011 trial criteria. However, as radiotherapy is deemed to have a major role in locoregional control, one concern is the radiation dose distribution in the axilla in “chest” versus “breast” irradiation. When measuring the incidental dose in the axilla, the radiation therapy technique needs to be taken into account. Indeed, breast and chest-wall irradiation has long been based on standard tangential fields and 3D conformal radiotherapy (3D-CRT). In a systematic literature review on axilla incidental radiation dose in patients treated with BCS, Schmitt et al. demonstrated that by using standard tangential fields and 3D conformal radiotherapy (3D-CRT), the average dose delivered to the axilla level I ranged between 20 and 43.5 Gy [10]. However, intensity-modulated radiotherapy (IMRT), delivering a more conformal dose distribution with a better coverage of target volumes, has been used in the treatment of breast cancer for 10 years, with the advantage of better sparing organs at risk [11]. Nevertheless, it increases low doses to healthy organs, with the risk of long-term side effects such as radiation-induced cancers. The total incidental axillary dose is technique-dependent and seems to decrease with IMRT, with considerable variability reported between authors, ranging from 14.5 to 42.6 Gy [10]. Some authors have reported a fixed-beam IMRT technique; others have reported a volumetric arc therapy (VMAT) technique, which consists of a rotational IMRT with the use of treatment arcs. “Tangential-like” VMAT is a technique with opposed small arcs that reproduces tangential irradiation.

In this sub-study, we attempted to assess the incidental axillary dose delivered by either a 3D-CRT technique or a VMAT technique in the setting of post-mastectomy radiotherapy within a subset population of a randomized controlled SERC trial [12]. This multi-centric trial, whose results are pending, aimed to confirm the results of the ACOSOG Z0011 through extending the patient population to those treated with a mastectomy or neo-adjuvant therapy, those with multi-centric tumors, those with extra-nodal extension, and those with any node involvement up to two macrometastases.

## 2. Materials and Methods

### 2.1. Study Population

Between July 2012 and September 2021, 2216 patients with T1-3 N0 breast cancer were included in a multicentric randomized phase III SERC trial (NCT01717131) described in a previous publication [12]. This trial was designed to assess the possibility of ALND avoidance in positive SLNB patients, extending the ACOSOG Z0011 patient population to those treated with a mastectomy, those treated with neo-adjuvant chemotherapy, and those with multicentric tumors and/or extracapsular extension. Briefly, after surgery and SLNB, patients with ≥1 macrometastases (and only ≥3 positive SLN or ≥1 and a mastectomy after amendment) were randomly assigned to further ALND or observation. Adjuvant treatments were administered according to the standard of care. In this sub-study, we selected patients treated with a mastectomy and PMRT in a single institution. We excluded patients with an indication of radiotherapy to the Berg level 1 (a rarely encountered indication at the discretion of the radiotherapist in cases of massive lymph node involvement after ALND).

### 2.2. Radiation Therapy

Post-mastectomy radiotherapy (PMRT) was recommended in patients with macroscopically positive lymph nodes. In the case of micrometastases or isolated tumor cells, PMRT was administered at the discretion of the medical team and based on other negative prognostic factors (young age, triple-negative or Her2+++ phenotype, and deep internal location). The radiation oncologist, according to the RTOG contouring guidelines, delineated both the target and organs at risk [13]. The Berg level 1 was delineated in all patients as required by the SERC protocol. Supra and infraclavicular fossa (Berg level 4 and level 3) were always included, while the retro-pectoral axillary area (Berg level 2 and inter-pectoral level) and internal mammary chain (IMC) were optional. Irradiation volumes were independent of the treatment arm.

Radiation delivery could involve one of the two following techniques at the radiation therapist’s discretion (also depending on evolving routine practices): (1) a mono-isocentric field-in-field 3D conformal radiotherapy (3D-CRT), (2) a volumetric arc therapy (VMAT) technique using a double 230° arc (170 to 300° for the left-side chest wall; 190 to 60° for the right-side chest wall—international standard IEC 61217 [14]). In the SLNB arm, high tangential fields were not permitted for the 3D-CRT technique. During the treatment planning, no objectives of coverage or constraint were given to the Berg level 1 volume. The prescription dose for the chest wall was 50 Gy in 25 fractions according to the SERC protocol. For the regional nodes, the most common prescription was 46 to 47 Gy in 23–25 fractions, depending on the technique used (46 Gy in 23 fractions of 2 Gy for 3D-CRT and 47 Gy in 25 fractions of 1.88 Gy for VMAT). Dose constraints on the organs at risk were as follows: volume receiving 20 Gy < 30% and volume receiving 30 Gy < 20% for the ipsilateral lung; volume receiving 5 Gy < 50% for both lungs; and mean heart dose < 5 Gy.

### 2.3. Outcomes

The primary objective of this sub-study was to determine the incidental dose delivered to the axillary area, defined as the Berg level 1. The secondary objective was to compare the dose distribution according to the techniques of radiotherapy (3D-CRT or VMAT).

### 2.4. Statistics

We used descriptive statistics for population characteristics (number, percentage, minimum, and maximum). We extracted the following dosimetric parameters for the Berg level 1 volume from dose-volume histograms and described them with average and standard deviation: mean dose (Dmean), V95%, D95%, D50%, and V40Gy. The first four parameters are classic dose reporting parameters. We also evaluated the volume receiving the dose of 40 Gy, as it can be considered clinically relevant for micrometastatic disease. We used a paired Student’s *t*-test to compare the two techniques. Significance was set at *p* < 0.05. We used R-software 4.3.2 version to compute statistical analyses.

This project was carried out within the SERC trial, which was approved by a local ethics committee.

## 3. Results

Over nine years, 439 of the 2216 patients included in the SERC trial underwent a mastectomy. Fifty-two patients treated in our institution had both a mastectomy and radiotherapy and met the inclusion criteria of this sub-study.

Patient characteristics are detailed in Table 1. Six patients with no palpable tumors underwent a mastectomy because of multicentric lesions or extensive in situ lesions. Four patients with pathological nodal stage N0 received PMRT: one had a tumor larger than 5 cm; two had isolated tumor cells in the node and associated risk factors (location, grade); and one had isolated tumor cells in the node and extensive in situ lesions. Patients treated using the VMAT technique were older and more likely to have left breast cancer. In addition, they had a higher number of positive lymph nodes, and histologic characteristics were more likely to be lobular subtype and hormone receptor-negative. The prescription dose was 50 Gy to the chest wall for all patients. Only one patient received treatment on the chest wall alone. She had an intraductal carcinoma extending over 95 mm with isolated tumor cells on the SLNB. The prescription dose to the lymph nodes was 46–47 Gy, depending on the technique. One patient in the VMAT group received 50 Gy in the node areas because she was included in a specific radiotherapy trial requiring this dose level. One patient of the 3D-CRT group received 50 Gy on the internal mammary chain area and 46 Gy on the Berg level 2, level 3, and level 4 due to her anatomy. Two patients were not irradiated on their Berg level 2 and level 3 (axillary retro pectoralis and infraclavicular). One had only isolated tumor cells with a pT3 tumor. The second had only micrometastases in the non-sentinel nodes at ALND after neoadjuvant chemotherapy, but the SLNB before chemotherapy involved two macrometastases. In the VMAT group, all axillary levels except the Berg level 1 were treated in all the patients.

The incidental dose delivered to the Berg level 1 is described in Table 2. The average incidental dose was 37.2 ± 8.5 Gy in the entire population. More than half of the axillary volume was covered by 95% of the prescribed dose (V95%). The volume receiving 40 Gy (V40Gy) was 67 ± 21.6% in the entire population. All the dosimetric parameters studied showed a significantly higher incidental dose delivered to the Berg level 1 in the VMAT group, compared with the 3D-CRT group. The Dmean was 43.6 Gy and 34.8 Gy in VMAT and 3D-CRT, respectively; *p* < 0.001. V40Gy was 80.4% versus 55.5% in the VMAT and 3D-CRT groups, respectively; *p* < 0.001. The mean dose delivered to the Berg level 1 for the patient treated with 3D-CRT on the chest wall alone was 22.9 Gy. The two patients in whom level 2 and level 3 were not treated were in the 3D-CRT group and received a Dmean of 26.7 and 31.7 Gy to the Berg level 1.

The dose distribution in the axilla is illustrated in Figure 1. This figure shows a visual comparison of the two techniques in the two parts of the axillary area constituting the Berg level 1: above and below the axillary vein. Dose homogeneity was greater in the VMAT group with better conformation. A more gradual decrease can be seen, with a preferential path for the dose into the axillary fossa with the VMAT technique. Above the axillary vein, the 40 Gy isodose (represented by the turquoise line) better covers the Berg level 1 in VMAT than in 3D-CRT.

## 4. Discussion

Our sub-study shows that post-mastectomy radiotherapy (PMRT) delivers a significant incidental axillary dose (AID). This AID is technique-dependent and is significantly higher with a volumetric arc therapy (VMAT) technique. The impact of the AID in terms of loco-regional control is not well established. Indeed, prospective studies so far have not correlated the AID to the rate of axillary recurrence. Nevertheless, it remains an important avenue of exploration, especially in the area of the de-escalation of both chemotherapy (in light of genomic test results) and axillary surgery.

The published data concerning this incidental axillary dose vary widely; Berg level 1 coverage is described as between 30 and 50% for 95% of the prescribed dose with standard tangential beams and between 1 and 39% with heterogeneous IMRT techniques. Some authors describe this coverage as inadequate [10]. However, axillary recurrence rates are thought to decrease after whole-breast radiotherapy under conservative treatment, which would explain the low rate of axillary recurrence in axillary lymph node dissection omission (ALND) trials [4,15,16]. In the context of PMRT, little is known about the AID due to scarce literature data. In the specific setting of immediate breast reconstruction with expanders and using tangential beams, Russo et al. reported a 23.9% coverage of Berg level 1 and 2 by 95% of the prescribed dose [17]. In our sub-study, we report a better coverage of the Berg level 1: 54.6% received at least 95% of the prescribed dose of radiotherapy to the chest wall and regional lymph nodes after a mastectomy without reconstruction. We report here the same range of the dose received by the Berg level 1 as published data after conservative surgery: from 22 to 43.5 Gy compared to 37.2 Gy in our sub-study [10].

These results may be useful to help decision-making and reduce the need for completion of ALND in pN1a patients treated with a mastectomy in relation to what is observed in a setting of radiotherapy after breast-conserving surgery (BCS). If Berg level 1 dose coverage is similar after both a mastectomy and BCS, there may no longer be any reason to complete axillary surgery in patients meeting the ACOSOG Z0011 trial criteria (except for the surgery type). Furthermore, the SINODAR-ONE trial, including patients with T1-T2 breast cancer with one or two macro-metastatic sentinel lymph nodes randomly assigned to further ALND or observation, found no difference in overall survival and recurrence-free-survival after 34 months of median follow-up between the two groups [18]. A sub-analysis of this trial regarding 218 patients treated with a mastectomy found sentinel lymph node biopsy (SLNB) alone to be non-inferior to further ALND, with no axillary lymph node recurrence. However, only 8% of the patients received PMRT in the SLNB arm compared to 27% in the ALND arm, which calls for an updating of the long-term outcomes in this population [19]. Three other prospective, randomized trials, whose results are pending, will soon shed light on this topic [20,21,22]. The SENOMAC trial, recently presented in an abstract form with some pre-specified secondary outcomes, investigated ALND omission in cT1-3N0 patients (whatever the surgery type) with up to two macrometastases after SLNB. It reported a low rate of axillary recurrence of 0.5% after a median follow-up of 37 months; 83.3% of patients received radiotherapy, including regional node irradiation. The results for the primary outcome as the rate of overall survival are awaited. The French SERC trial also followed the ACOSOG Z0011 trial design, extending the inclusion criteria to patients with cT1-T2 and pT3 tumors, mastectomy, extracapsular invasion, or any positive sentinel lymph node (SLN) up to two macrometastases [20]. The publication of disease-free survival data is awaited and could reinforce the first SENOMAC data. In the POSNOC trial, breast cancer patients with unifocal or multifocal cT1-2 invasive tumors, with one or two SLN macrometastases and treated with either BCS or a mastectomy, were randomly assigned to adjuvant therapy alone (endocrine therapy, chemotherapy, and/or radiotherapy) or adjuvant therapy plus axillary treatment (surgery or radiotherapy). The primary outcome is 5 years for axillary recurrence.

Many studies have reported the AID after BCS, with highly variable data depending on the technique used [10]. The increasing use of the IMRT technique in breast cancer radiotherapy is a source of concern due to the risk of axillary underdosing. Several studies have described poorer coverage of the axillary region due to a more conformal dose distribution with the use of IMRT [23,24,25]. Two of these studies used five to seven fixed beams and obtained a level 1 mean dose of 14.5 Gy and 27.7 Gy, respectively [23,24]. Two studies using tangents IMRT (or Forward IMRT) found a level 1 mean dose of 29.1 Gy (field-in-field technique) and 39 Gy, respectively [24,25]. More recently, Ahrouch et al. compared the AID according to the radiotherapy technique—Forward IMRT, 3D-CRT, VMAT (1 × 270°), and “tangential-like” VMAT (two treatment arcs less than 90° in angle)—in breast irradiation alone after BCS [26]. They found no significant difference in the dose delivered to the Berg level 1 among all techniques. However, the distribution of low doses within the axillary area was greater in the VMAT group (V20Gy). The mean dose in the Berg level 1 was slightly higher when using the “tangential-like” VMAT (2 × 90°) in comparison to a 270° single-arc VMAT—34 Gy and 31.8 Gy, respectively. In our sub-study, the use of a 230° double-arc VMAT showed that the average dose in the Berg level 1 was 43.6 Gy.

However, in our sub-study, regional nodes were part of the target volume. Other data show that the technique has a major impact on the AID. Two studies found deep inspiration breath hold (after either BCS or a mastectomy) to significantly reduce the Dmean dose in the Berg level 1 by around 3 Gy as compared with free-breathing radiotherapy [27,28]. For this reason, it could be suggested that the axillary dose, even if incidental, should be systematically recorded to correlate its relationship with axillary control, especially in the case of ALND omission after positive SLNB, whatever the type of surgery.

Another point is the systematic axillary treatment of the Berg level 2 in our sub-study (except for three patients). This strategy follows current treatment recommendations but was not systematic in the past when the 3D-CRT technique was mainly used. Thus, our study provides new data on PMRT and the contribution of “modern” lymph node irradiation. Most of the studies on incidental axillary irradiation consider breast radiotherapy without lymph node irradiation. One study including lymph node irradiation found an incidental mean dose to the Berg level 1 of 38.2 Gy, similar to our results (34.8 Gy with a 3D-CRT technique) [27]. In our sub-study, all the patients in the VMAT group received irradiation to their Berg level 2 and level 3, as recommended in lymph node macrometastases. This may explain the good coverage of the Berg level 1, specifically in the area above the axillary vein (Figure 1). Investigating radiotherapists in the SERC trial had to comply with RTOG contouring guidelines, including for the Berg level 2 region. These follow the axillary vein in relation to the pectoralis minor muscle, and the posterior limit corresponds to the ribs. Other international recommendations are also based on the pectoralis minor muscle but introduce the notion of the upper limit of ALND when marked by a clip [13]. For clinical practice, the use of a clip to delimit the upper part of the ALND, and therefore the limit of radiotherapy fields, seems entirely appropriate.

Our study has its strengths and limitations. On the one hand, its strengths are that data came from a randomized clinical trial where radiotherapy treatment was standardized. Volume contouring followed international recommendations. To avoid voluntary irradiation or avoidance of the axillary area, no dose constraints were applied to the Berg level 1 region during inverse planning in the VMAT technique. However, the use of inverse planning likely explains the good coverage of the Berg level 1 region in the VMAT group. Indeed, the Treatment Planning System tends to protect organs at risk and cover the volume to be treated, and the axillary area remains a preferential path for the dose, with no constraints nor targets at this level. Of note, the same organ-at-risk constraints were used for all patients, whatever the radiotherapy technique. For the 3D-CRT technique, the use of a “high tangential” technique (cranial tangent border 2 cm away from the humeral head) was not authorized in the SERC trial. Indeed, some authors have described the axillary dose distribution of the high tangential technique as equivalent to voluntary axillary irradiation, as in the AMAROS trial [29,30,31]. High tangential techniques were used in half of the patients in the ACOSOG Z0011 trial, which is considered biased in the interpretation of the axillary recurrence rate [32]. In the SERC trial, the investigators wanted to ensure that the irradiation of the axillary area concerned with SLNB remained involuntary. However, it is interesting to note that in the context of PMRT, the RTOG recommendations used for delineation determine the upper part of the volume of the chest wall as being the lower part of the sternoclavicular joint. This upper limit is not very far from the humeral head in terms of height. This is a point that may explain the good coverage of the upper part of the axillary area in our specific context of post-mastectomy radiotherapy. On the other hand, our study has limits. Although carried out as part of a prospective randomized trial, this study was performed retrospectively and not planned at baseline. Only patients treated in one radiotherapy center of the SERC trial were included because radiotherapy data were partially collected for the main study, and it was necessary to have access to the radiotherapy plan to collect all the dosimetric parameters. There were some differences between our two groups of treatment, divided according to the technique used, which was left to the practitioner’s choice. First, patients in the VMAT group were older. Indeed, due to the risk of cancer induced by low-dose radiation, patients under the age of 50 were preferentially treated with 3D-CRT. There were more patients treated for left-side breast cancer in the VMAT group, probably because of better cardiac sparing with the use of this technique. There were more hormone receptor-negative and lobular tumors in the VMAT group. However, these items should not have affected dosimetry. One difference between the two groups could have affected the dosimetry: the three patients for whom the Berg level 2 and level 3 regions were not treated were in the 3D-CRT group. Finally, despite the small number of patients, we were able to show a significant difference between the techniques, probably thanks to the homogeneity of the practices in a single center.

In the AMAROS trial, voluntary axillary radiotherapy was non-inferior to ALND for axillary recurrence, but there are currently no data on incidental versus voluntary axillary irradiation in terms of either axillary recurrence or toxicity [6]. The extension of nodal irradiation to the axillary area is suspected to increase the risk of lymphedema [33]. However, Warren et al. found that tangential radiotherapy with its incidental axillary dose did not increase the rate of lymphedema, which remained at 3% with or without irradiation of the breast or chest wall alone. The addition of supra and infraclavicular fossa irradiation was sufficient to increase the risk of lymphedema, estimated at 21.6% [34]. When adding a posterior axillary beam, the lymphedema rate was similar, with 21.1%. On the other hand, another more recent retrospective trial reviewed 1369 patients with pT1-3N0-1 breast cancer who underwent adjuvant radiotherapy. There were no significant differences for lymphedema after the adjunction of regional node irradiation after SLNB versus SLNB alone, despite 70% of axillary radiotherapy (including the Berg level 1 region) in the regional node irradiation group [35]. Moreover, there is a risk of increasing lung irradiation, with additional potential risk of a second cancer. In the recently published 10-year data from the AMAROS trial, the authors found more second cancers in the axillary radiotherapy group: 75 events (12.1%) in the axillary radiotherapy group versus 57 (8.3%) in the ALND group. Of these events, 21 in the axillary radiotherapy group had contralateral breast cancer, which did not appear to be related to the addition of axillary radiotherapy, compared with 11 in the ALND group. The published data do not give the number of lung cancers in the two groups, but there were seven deaths due to lung cancer in the axillary radiotherapy group compared with five in the ALND group [6]. All these data come from 3D radiotherapy techniques; therefore, we need more data on the toxicity of modern radiotherapy techniques like VMAT. With the most recent techniques, the toxicity of axillary radiotherapy, whether voluntary or incidental, may not be so different.

## 5. Conclusions

Our sub-study shows a significant incidental axillary dose after mastectomy that is at least as high as after conservative treatment in breast cancer patients. These results suggest that axillary recurrence may vary mainly according to whether or not radiotherapy is added, independently of the type of breast surgery. This could influence future therapeutic decisions regarding post-mastectomy axillary treatment. At present, the difference between the voluntary addition of axillary radiotherapy or involuntary irradiation in terms of toxicity and axillary control is unknown. We need clinical data to answer this question and to take modern radiotherapy techniques into account. Due to the variability of this incidental axillary dose, it could be interesting to record the dose administered to the Berg level 1 region in clinical practice, even when it is not deliberately treated.

## Figures and Tables

**Figure 1 cancers-16-01198-f001:**
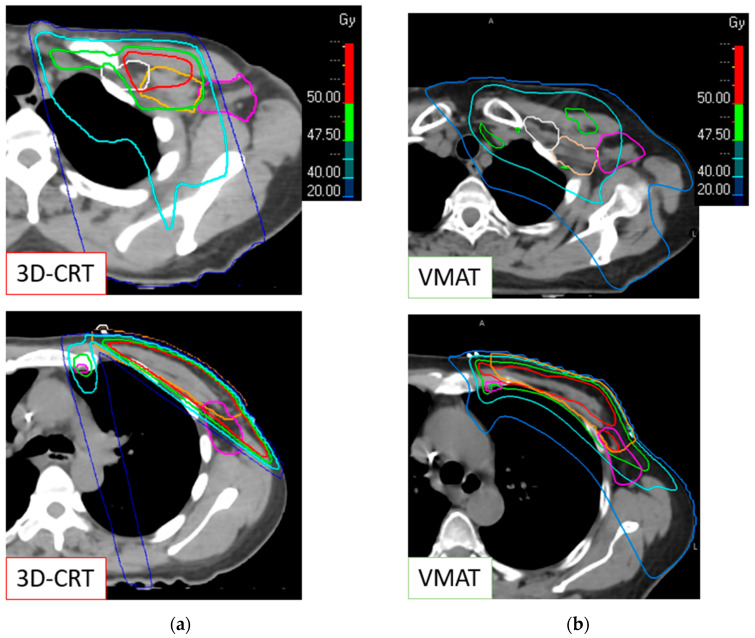
Incidental axillary dose distribution, with a focus on the upper and lower part of the Berg level 1. (**a**) With a 3D conformal radiotherapy therapy (3D-CRT) technique; (**b**) with a volumetric arc therapy (VMAT) technique. Pink line: Berg level 1; orange line: Berg level 2; white line: Berg level 3.

**Table 1 cancers-16-01198-t001:** Clinical, pathological, and treatment characteristics of the population.

	Total (*n* = 52)	3D-CRT (*n* = 28)	VMAT (*n* = 24)
Age (mean (range))	55 (32–79)	53 (33–77)	57 (32–79)
Side - Right - Left	28 (53.85) 24 (46.15)	16 (57.14) 12 (42.86)	12 (50.00) 12 (50.00)
Clinical Tumor Stage - T0 - T1 - T2 - T3	6 (11.54) 12 (23.08) 33 (63.46) 1 (1.92)	3 (10.71) 5 (17.86) 20 (71.43) -	3 (12.50) 7 (29.17) 13 (54.17) 1 (4.17)
Tumor Size (mm) (mean (range))	27.73 (0–150.00)	30.33 (4.00–150.00)	25.49 (0–65.00)
Pathological Tumor Stage - ypT0 - ypTis - pT1 - pT2 - pT3	1 (1.92) 1 (1.92) 17 (32.69) 29 (55.77) 4 (7.69)	1 (3.57) 1 (3.57) 10 (35.71) 13 (46.43) 3 (10.71)	- - 7 (29.17) 16 (66.67) 1 (4.17)
Pathological Nodal Stage - pN0(i+) - pN1mi - pN1 - pN2a	4 (7.69) 8 (15.38) 38 (73.08) 2 (3.85)	3 (10.71) 4 (14.29) 19 (67.86) 2 (7.14)	1 (4.17) 4 (16.67) 19 (77.19) -
Histology type - Infiltrating ductal - Infiltrating lobular - Mixed - Others	37 (71.15) 11 (21.15) 2 (3.85) 2 (3.85)	23 (82.14) 2 (7.14) 1 (3.57) 2 (7.14)	14 (58.33) 9 (37.50) 1 (4.17) -
SBR Grade - I - II - III	14 (26.92) 21 (40.38) 17 (32.69)	8 (28.57) 10 (35.71) 10 (35.71)	6 (25) 11 (45.83) 7 (29.17)
LVI - Yes - No - Unknown	20 (38.46) 29 (55.77) 3 (5.77)	11 (39.29) 15 (53.57) 2 (7.14)	9 (37.50) 14 (58.33) 1 (4.17)
Endocrine receptors status - Positive - Negative - Unknown	44 (84.62) 7 (13.46) 1 (1.92)	25 (89.29) 2 (7.14) 1 (3.57)	19 (79.2) 5 (20.8) -
HER2 status - Positive - Negative - Unknown	9 (17.31) 42 (80.77) 1 (1.92)	5 (17.86) 22 (78.57) 1 (3.57)	4 (16.67) 20 (83.33) -
cALND	22 (42.31)	12 (42.86)	10 (41.67)
NSN+ ^$^	5 (22.73)	2 (16.67)	3 (30)
Extracapsular invasion - Yes - No - Unknown	13 (25) 38 (73.08) 1 (1.92)	4 (14.29) 23 (82.14) 1 (3.57)	9 (37.5) 15 (62.5) -
Lymph node area target - Berg 4 (supra-clavicular) - Berg 3 (infra-clavicular) - Berg 2 (retropectoral) - IMC	51 (98) 49 (94) 49 (94) 51 (98)	27 (96) 25 (89) 25 (89) 27 (96)	24 (100) 24 (100) 24 (100) 24 (100)

^$:^ The percentage for this item is related to the number of patients who received completion of ALND. Abbreviations: cALND: completion axillary lymph node dissection, IMC: internal mammary chain, 3D-CRT: three-dimensional conformal radiation therapy, VMAT: volumetric modulated arc therapy, NSN: non-sentinel Node, LVI: lymphovascular invasion, SBR: Scarff–Bloom–Richardson.

**Table 2 cancers-16-01198-t002:** Comparison of incidental dose at the Berg level 1 between conformal 3D radiotherapy and volumetric arc therapy.

	Total (*n* = 52)	3D-CRT (*n* = 28)	VMAT (*n* = 24)	*p*-Value *t*-Test
Dmean	37.2 Gy (SD = 8.5 Gy)	34.8 Gy (SD = 8.6 Gy)	43.6 Gy (SD = 3.1 Gy)	<0.001
V95%	54.6% (SD = 19.0%)	49.4% (SD = 20.7%)	60.8% (SD = 15.2%)	0.027
V40Gy	67.0% (SD = 21.6%)	55.5% (SD = 22.3%)	80.4% (SD = 10.3%)	<0.001
D50%	41.5 Gy (SD = 9.9 Gy)	37.8 Gy (SD = 12.3 Gy)	45.8 Gy (SD = 2.1 Gy)	0.002
D95%	18.6 Gy (SD = 13.8 Gy)	7.3 Gy (SD = 6.7 Gy)	31.8 Gy (SD = 5.8 Gy)	<0.001

Abbreviations: 3D-CRT: 3D conformal radiotherapy; VMAT: volumetric arc therapy; SD: standard deviation; Dmean: mean dose; V95%: volume receiving 95% of the prescribed dose; V40Gy: volume receiving 40 Gy; D50% dose received by 50% of the volume; D95%: dose received by 95% of the volume.

## Data Availability

Data are contained within the article.

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
