# Peer review of "Does Breast Surgery Type Alter Incidental Axillary Irradiation? A Dosimetric Analysis of the “Sentinel Envahi et Randomisation du Curage” SERC Trial"

_cancers, 2024, doi:10.3390/cancers16061198_

Round 1
Reviewer 1 Report
Comments and Suggestions for Authors
I have appreciated the manuscript, in which you retrospectively examined the incidental dose to the axilla (Berg level 1) in a sub-group of patients of the SERC trial treated with total mastectomy.
Several studies examined the incidental dose to the axilla, with respect to different radiation therapy (RT) techniques, but generally the investigation was limited to patients treated after breast conserving surgery.
The main strengths of this analysis are:
- - the inclusion only of patients who underwent total mastectomy
- - the standardization of the RT treatment, according to SERC protocol, and RTOG contouring guidelines.
- - the comparison of two different techniques, three-dimensional conformal RT (3D-CRT) and volumetric arc-therapy (VMAT).
There are some weaknesses, however, as mentioned in the Discussion:
- it is a retrospective study, even though patients had been enrolled in the multi-centric randomized phase III SERC trial;
- the number of cases included in the analysis is limited (52 patients) and the study findings cannot cover the great variability of the incidental dose to L1.
A longer follow-up and a larger number of patients, from all the Centers of the SERC trial, are needed to confirm the results of this study. Besides the outcome in terms of loco-regional control and toxicity should be reported in a future paper.
The text is clear and exhaustive, the results are critically discussed, the Tables properly illustrate the data and the Figure clearly shows dose distribution.
On the whole I think that this is an interesting study, that deserves consideration for publication.
Author Response
Dear reviewer,
Thank you for your review and attention to our manuscript on axillary incidental dose after total mastectomy.
As you pointed out, this is a subgroup studied retrospectively in a single center of the SERC trial with a limited number of patients. As explained in the discussion, the need to analyze the dose-volume histograms of each patient forced us to limit ourselves to a single center. As you suggested, it could be very enriching in a future study to report some radiotherapy data from all SERC trial centers and try to correlate them with locoregional control and toxicity data.
Please find attached the revised manuscript with the visible changes.
Reviewer 2 Report
Comments and Suggestions for Authors
It is a pleasure to review this study, entitled "Does breast surgery type alter incidental axillary irradiation? Dosimetric analysis of the SERC trial," which addresses a significant gap in breast cancer treatment literature by analyzing the incidental axillary dose (AID) delivered during post-mastectomy radiotherapy (PMRT) using different radiotherapy techniques. The manuscript provides a comprehensive examination of how the choice of surgical procedure, specifically breast conserving surgery (BCS) versus mastectomy, influences the incidental irradiation of the axillary region, which has implications for axillary recurrence rates in breast cancer patients.
The authors utilized data from a subset of patients within the multicentric randomized phase III SERC trial, focusing on those who underwent mastectomy followed by PMRT, comparing the dosimetric outcomes between 3D-conformational radiotherapy (3D-CRT) and volumetric arc therapy (VMAT). Their findings reveal a significant difference in AID between the two techniques, with VMAT delivering a higher incidental dose to the axillary area defined as Berg level 1. This higher dose could potentially influence loco-regional control and, by extension, the need for additional axillary treatments in certain patient groups.
The manuscript is methodologically sound, well-written, and provides a valuable contribution to the field by suggesting that the type of radiotherapy technique used post-mastectomy can impact the incidental dose to the axilla, which might influence treatment decisions for patients with breast cancer. The study is timely, given the current trends towards treatment de-escalation, and adds important data to the ongoing discussion about optimizing breast cancer treatment while minimizing adverse effects.
The positive review is well-deserved, and I think it can be accepted in the present form
Author Response
Dear reviewer,
Thank you for your time and your interest in our manuscript and for your positive feedback.
Please find attached the revised manuscript with the visible changes.
Reviewer 3 Report
Comments and Suggestions for Authors
From my point of view, this is a very nice radiotherapeutic work that deals with an important radio-oncological topic. The results are clearly presented, explained and discussed. In my opinion, however, one thing should be emphasized a little more: A key aspect of the analysis is the fact that in the substudy the regional nodes were part of the target volume, which is after all probably one reason why even a higher dose could be achieved with VMAT. What do the authors specifically recommend with regard to the target volume definition? Such a paragraph for the readers' clinical practice would be interesting.
Author Response
Dear reviewer,
Thank you for your time and your positive feedback on our work.
To answer your question about target volumes definition, this is indeed an important point in our study. One of the strengths of the analysis is that target volumes were homogeneous within the SERC trial. As explained in our methods (l.136), the investigating radiotherapists in the SERC trial were required to comply with RTOG guidelines for target and organ-at-risk delineation, including lymph node volumes, which made it possible to analyze patient’s dosimetry with homogeneous target volumes. Your comment shows us that this explanation is not sufficient for the reader's understanding and practice, so we have added a reference for the RTOG contouring guidelines (l.137, quotation 13). We've also added a sentence to the discussion paragraph (l.314-320) on the link between volume of lymph node areas, particularly axillary berg level 2, and the good coverage of berg level 1 with the VMAT technique :
“Investigating radiotherapists in the SERC trial had to comply with RTOG contouring guidelines, including for Berg level 2. These follow the axillary vein in relation to the pectoralis minor muscle, and the posterior limit corresponds to the ribs. Other international recommendations are also based on the pectoralis minor muscle, but introduce the notion of the upper limit of ALND when marked by a clip [13]. For clinical practice, the use of a clip to delimit the upper part of the ALND, and therefore the limit of the radiotherapy fields, seems entirely appropriate. »
Please find attached the revised manuscript with the visible changes.
Reviewer 4 Report
Comments and Suggestions for Authors
Congratulations. The scientific paper "Does breast surgery type alter incidental axillary irradiation? Dosimetric analysis of the SERC trial" represents another step forward in the ongoing search for the appropriate adjuvant treatment in the most frequently met type of cancer. The authors successfully attempted a SERC sub-study to assess the axillary incidental dose delivered by either a 3D-CRT technique or a VMAT technique for post-mastectomy patients. It is a great work with practice changing potential in the foreseeable future. Apparently the SERC trial still has a lot to show. As it was the case with incidental axillary radiation after BCS, these patients also received a significant radiation dose to Berg level I, despite it was not included in the target volume. The authors' conclusion that the PMRT could justify the low axillary relapse rate seen in other studies independently of the type of breast surgery, receives a lot of credit.
Minor English language issues:
- line 29 "affects" not "affect"
- line 30: "completing" instead of "completion"...and a few others. Please comb the text again thoroughly for English language.
Comments on the Quality of English LanguageEnglish language needs minor fixes as highlighted above.
Author Response
Dear reviewer,
Thank you for your positive feedback and your enthusiasm about our sub-study. Thank you for your comments regarding the English language. We have taken your comments into account and modified the text accordingly.
Please find attached the revised manuscript with the visible changes.